# Identification of immunodominant linear epitopes from SARS-CoV-2 patient plasma

**Lluc Farrera-Soler[1], Jean-Pierre Daguer[1], Sofia Barluenga[1], Oscar Vadas[2], Patrick Cohen[3], Sabrina Pagano[3], Sabine Yerly[3], Laurent Kaiser[3,4], Nicolas Vuilleumier[3], Nicolas Winssinger[1]***

**1** Department of Organic Chemistry, NCCR Chemical Biology, Faculty of Science, University of Geneva, Geneva, Switzerland, **2** Department of Microbiology and Molecular Medicine, University of Geneva, Geneva, Switzerland, **3** Division of Laboratory Medicine, Diagnostic Department, Geneva University Hospitals and Faculty of Medicine, Geneva, Switzerland, **4** Division of Infectious Diseases¸ Geneva University Hospitals and Faculty of Medicine, Geneva, Switzerland

* Nicolas.Winssinger@unige.ch

## Abstract

A novel severe acute respiratory syndrome coronavirus (SARS-CoV-2) is the source of a current pandemic (COVID-19) with devastating consequences in public health and economic stability. Using a peptide array to map the antibody response of plasma from healing patients (12) and heathy patients (6), we identified three immunodominant linear epitopes, two of which correspond to key proteolytic sites on the spike protein (S1/S2 and S2') known to be critical for cellular entry. We show biochemical evidence that plasma positive for the epitope adjacent to the S1/S2 cleavage site inhibits furin-mediated proteolysis of spike.

## Introduction

On December 2019, a novel infectious disease causing pneumonia-like symptoms was identified in the city of Wuhan in the province of Hubei (China) [1]. This new coronavirus infectious disease (COVID-19) caused by the severe acute respiratory syndrome coronavirus-2 (SARS-CoV-2) is having a devastating impact on public health and economic stability on a global scale [2]. The World Health Organization declared it a pandemic on the 11th March 2020.

Mapping the epitopes corresponding to the immune system's antibody response against the virus is important for vaccine development [3, 4], diagnostic serological tests [4] as well as for identifying neutralizing antibodies with therapeutic potential [5]. Indeed, epitope mapping of the SARS-CoV-1 revealed immunodominant epitopes and identified neutralizing antibodies [6–13]. However, the observation of antibody-dependent enhancement (ADE) of SARS-CoV-1 in non-human primates is concerning and should be considered for vaccine development [14, 15]. While ADE mechanisms arising from binding-only antibodies (non-neutralizing) are well documented, an ADE mechanism with neutralizing antibodies for the related MERS-CoV was also reported [16]. In this case, it was shown that neutralizing antibodies targeting the receptor-binding domain (RBD) of the virus redirected viral entry to Fc-expressing cells, broadening the host-targeted cells. Thus, antibodies generated by vaccination against

**Funding:** The work was funded by research funds from the University of Geneva and the département d'instruction public (DIP) du canton de Genève.

**Competing interests:** The authors have declared that no competing interests exist.

SARS-CoV-2 could enhance viral entry instead of offering protection, leading to vaccine-associated enhanced respiratory disease (VARED) [17].

The homology between SARS-CoV-1 and SARS-CoV-2 rapidly led to the hypothesis that neutralizing antibodies identified from patients in the SARS-CoV-1 in the 2003 epidemic could also be neutralizing SARS-CoV-2 [18, 19]. Other antibodies with neutralizing activities have been discovered through different methodologies [20–25]. The rapid propagation of SARS-CoV-2 stimulated several studies predicting the antigenic parts of the viral proteins *in silico* [26–32], and analyzing SARS-CoV-1 epitopes that were conserved in this new coronavirus [33–36]. More recently, the first reports of experimental epitope mapping of the SARS-CoV-2 were deposited on repositories [37–42].

Herein we report the preparation of a microarray to map the antibody response to linear epitopes of the spike protein of SARS-CoV-2 and the analysis of 12 laboratory confirmed COVID-19 cases and 6 negative controls using the described peptide microarray.

## Materials and methods

### Plasma specimens from COVID-19 and healthy patients

Anonymized leftovers of whole blood-EDTA collected for routine diagnostic purposes under a general informed consent were used for this study, according to the Cantonal Research Ethics Commission of Geneva, Switzerland and Swiss regulations. In accordance with the article Number 2b of the Swiss law on human research regarding the use of anonymized biological material, no specific ethical approval was requested.

We included 12 real-time RT-PCR confirmed COVID-19 cases hospitalized at the University Hospitals of Geneva, and 6 unmatched negative blood samples from asymptomatic donors, obtained during the same period (April 2020). Analyses (see below) were performed within 72h of blood sampling without any freezing-thawing cycle.

**SARS-CoV-2 RT-PCR analyses and SARS-CoV-2 IgG serology.**   As previously published [43], SARS-CoV-2 RT-PCR was performed according to manufacturers' instructions on various platforms, including BD SARS-CoV-2 reagent kit for BD Max system (Becton, Dickinson and Co, US) and Cobas 6800 SARS-CoV-2 RT-PCR (Roche, Switzerland).

SARS-CoV-2 IgG serology against the S1-domain of the spike protein of SARS-CoV-2 was assessed using the CE-marked Euroimmun IgG ELISA (Euroimmun AG, Lübeck, Germany # EI 2606–9601 G). EDTA-plasma was diluted at 1:101 and assessed with the IgG ELISA according to the manufacturer's instructions and has been extensively reported elsewhere [43]. Median time from RT-PCR to serology testing was 3 weeks, reason why sample were considered as healing rather than convalescent plasma. All the 12 COVID-19 samples were considered as reactive against SARS-CoV-2.

### Synthesis of the peptide-PNA conjugate library

The library of peptide-PNA conjugate was synthesized by automated synthesis on an Intavis peptide synthesizer as previously described [44, 45]. The synthesis was initiated with the peptide followed by the PNA tag using a capping cycle after each coupling. Hence, truncated peptides cannot hybridize on the microarray since they will not have the necessary tag. A library of 200 linear peptides was constructed based on the sequences of the spike ectodomain protein from SARS-CoV-2 (residues 1-1213-Gene Bank: QHD43416.1), fragmenting the protein into two sets of 100 peptides (12*mer*) with an overlap of 6 residues. Each peptide-PNA conjugate was positively identified by MALDI analysis. See SI for full synthetic details and characterization data.

## Microarray epitope mapping

Microarrays were obtained from Agilent (Custom microarray slides, Agilent ref:0309317100–100002). Each peptide-PNA is complementary to a DNA sequence that is present 23 times at random positions on the array.

The arrays were incubated with plasma (1:150 dilution) for 1 hour at room temperature, washed with PBS-T and dried by centrifugation prior to the next step. The arrays were then incubated with Cy-3 labeled goat anti-human IgG (ab97170 from Abcam, 1:500 dilution) for 30 min, washed with PBS-T and dried by centrifugation for scanning. See SI for detailed procedures.

The fluorescence intensity on the array was measured on a GenePix 4100A microarray scanner using the median value of fluorescent intensity. The data for each peptide (23 spots) was plotted as a heat map of the median value from the 23 spots. The high redundancy in the measurements and the use of a median function insures that artifacts from a microarray experiment do not contribute to the consolidated data.

## Validation of epitope 655–672

The peptide was synthesized according to the same protocol as for the library synthesis, replacing the PNA tag with a biotin.

**Fluorescent bead assay.** 0.3 μL of Pierce™ High-Capacity Streptavidin Agarose Beads (catalog n°: 20357 from Thermo Scientific™) were mixed with 50 μL of the biotinylated peptide 10μM in PBS- T. The beads were incubated for 20 minutes and thereafter blocked with 200 μL of Fetal bovine serum for 10 minutes. The beads were then washed once with 100 μL PBS-T and 5 μL of serum from either positive or negative patients was added together with 450 μL of PBS-T and 50 μL of fetal bovine serum in order to block unspecific interactions. The beads were then incubated for 90 minutes and subsequently washed 4 times with 100 μL of PBS- T in order to remove all the non-binders. Finally, 200 μL of a 163nM solution of anti-human IgG-FITC (ab6854 from Abcam) in PBS-T with 0.5% BSA was added and incubated for 1 hour. The excess of secondary antibody was washed away using 3 times 100 μL of PBS-T and finally the beads were imaged with a Leica SP8 inverted confocal microscope.

**By Enzyme-Linked Immunosorbent Assay (ELISA).** A solution of Streptavidin (ref: S0677 from Sigma Aldrich), 100 μL of an 80nM, in PBS was added to a Corning® 96-well Clear Flat Bottom Polystyrene High Bind Microplate (catalog n°: 9018 from Corning) and incubated overnight at 4˚C. The plate was then washed three times with 300 μL of PBS-T (60 seconds, room temperature) and 200 μL of an 800nM solution of biotinylated peptide in PBS-T was added and incubated for 90 minutes at 36˚C. The plate was then blocked with 300 μL of PBS-T with 0.5% non-fat dry milk (60 minutes at 36˚C). The plate was washed 3 times with 300 μL of PBS-T (60 seconds, room temperature) and a 1:300 diluted plasma in PBS-T-0.5% non-fat dry milk was added to each well and incubated for 90 minutes at 36˚C. After incubation of the plasma, the plate was washed 3 times with 300 μL PBS-T (60 seconds, room temperature), 1 time with PBS-T 0.5% non-fat dry milk (60 minutes, 37˚C) and again 3 times with 300 μL PBS-T (60 seconds, room temperature). 100 μL of Goat Anti-Human-IgG HRP conjugated (ab97175 from Abcam) 1:10000 diluted in PBS-T 0.5% BSA were added to each well and incubated for 90 minutes at 37˚C. The plate was then washed 3 times with PBS-T (60 seconds, room temperature) and 200 μL of a 0.41mM solution of 3,3′,5,5′-Tetramethylbenzidine (TMB) (ref: 860336 from Sigma Aldrich) in 50mM $Na_2HPO_4$, 25mM citric acid and 0.0024% $H_2O_2$, pH 5.5 solution was added to the plate and incubated for 20 minutes at 37˚C. Finally, 50 μL of a 1M sulfuric acid solution were added and the absorbance was measured at 450nm with a plate reader (SpectroMax, Molecular Device). For each sample, triplicates were performed and the fluorescence value are the average of the 3 reads.

## Expression, purification and labeling of nCov-19 spike protein

Recombinant nCov-19 Spike protein ectodomain (16–1208) was subcloned into a modified pFastBac vector encoding a N-terminal Gp67 secretion signal. Following the nCov-19 Spike protein coding sequence, a C-terminal T4 fibritin trimerization domain, a TEV protease recognition site, a TwinStrepTag and a 10XHisTag were added. Baculovirus were generated following the Bac-to-Bac expression system. Two liters of Sf9 insect cells infected with baculoviruses expressing the protein of interest were infected for 70 hours, the media containing the secreted protein was clarified by a centrifugation at 4000 g for 15 min at 4˚C and the media was concentrated to 40 mL using tangential filtration (Vivaflow 200, 30 MWCO). Concentrated media was filtrated using 0.22 μm filter and loaded on a 5 mL Strep-Tactin XT Superflow column (IBA). Column was washed with 50 mL of Phosphate Buffer Saline (PBS) and 150 μg of TEV protease were added to the column. After 1 hour incubation at room temperature, the untagged protein was eluted using PBS. TEV protease was removed by applying the sample to a 1 mL His-Trap FF column. The protein was then injected onto a Superdex 200 10/300 column equilibrated in PBS at 4˚C. Fractions containing the trimerized nCov-19 Spike protein were pooled and concentrated before flash freezing in liquid nitrogen. For the labeling of the protein, 2 μL of Dylight 549 activated ester (1mg/mL in DMF, Thermo Scientific, ref: 46407) were added into 40 μL of Spike protein (0.35mg/mL in PBS) and incubated for 90 min at 23˚C. The unreacted dye was removed with Pierce™ Dye Removal Columns (Thermo Scientific, ref: 22858).

## Antibody-mediated inhibition of the furin-proteolysis of nCov-19 spike protein

10 μL of labeled spike protein (0.27 mg/mL) in $CaCl_2$ buffer (20mM Tris, 4mM $CaCl_2$, 100mM NaCl, pH 8.0) was mixed with 10 μL of serum and incubated for 1h at 22˚C. Following this incubation, 1μL of furin (recombinant human furin protein, R&D Systems, ref: 1503-SE-010) was added and the reaction incubated for another 45 min at 37˚C. Finally, the cleavage of the Spike protein was followed by SDS-PAGE and Cy3 scanning.

## Sequence alignment

Sequence alignment was done using Clustal Omega [46].

# Results and discussion

SARS-CoV-2 is composed of 4 major structural proteins: S (spike), M (membrane), N (nucleocapsid) and E (envelope) [47–49]. The spike protein is responsible for entry by binding the angiotensin-converting enzyme 2 (ACE 2) on the host cell [50, 51]. Accordingly, antibodies that bind the RBD and inhibit the interaction of the S protein with ACE 2 have been the center of attention. Based on the critical role of the S protein in CoV infection, we focused our work on this protein, dissecting it into two sets of overlapping linear 12*mer* peptides (two-fold sequence coverage with 6AA overlap between the two sets; i.e 1–12, 7–18, 13–24,. . .). The peptide array was prepared by hybridization of PNA-tagged peptide library onto a DNA microarray (Fig 1) [52]. This technology insures a high level of homogeneity across different arrays since individual arrays are prepared from the same library hybridized onto commercial DNA microarrays. Furthermore, the arrays are designed to have each sequence present 23 times, thus insuring high accuracy by calculating the median of the observed fluorescence of the 23 spots.

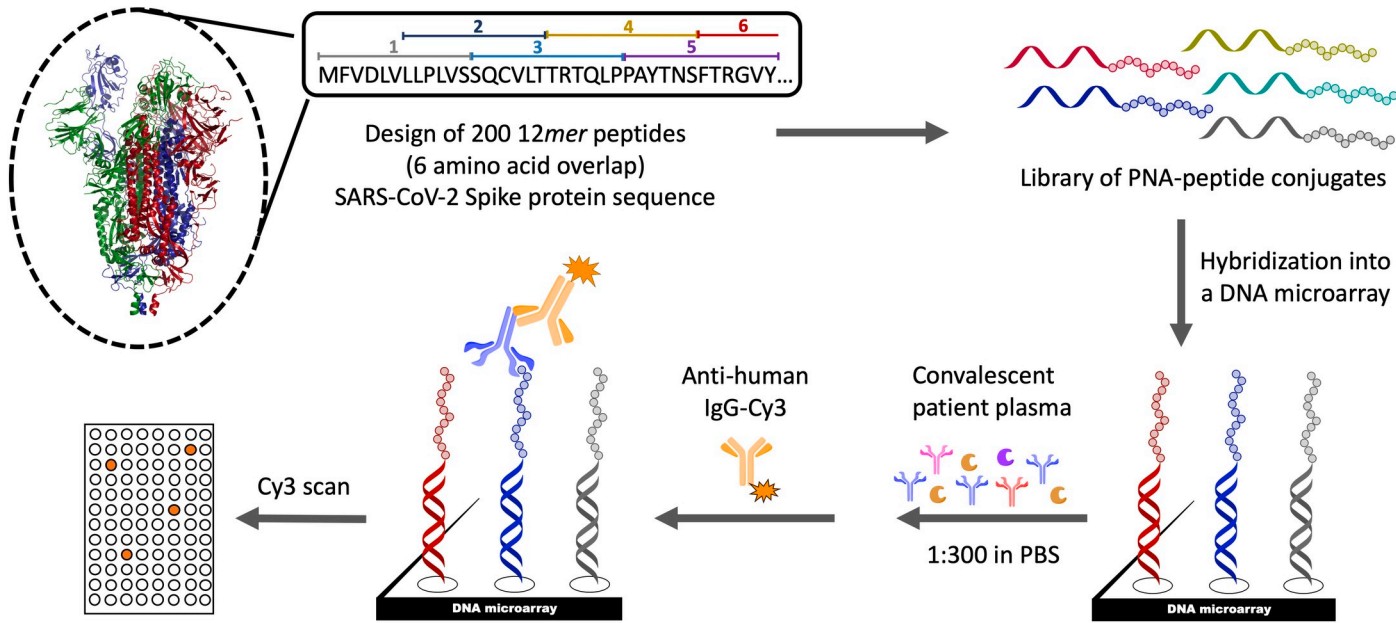

**Fig 1. Schematic representation of the 200 membered peptide-PNA epitope library design, DNA microarray generation and experimental approach to map the antibody response of COVID-19 patients to the S protein of SARS-CoV-2.**

The S protein of SARS-CoV-2 shares 76% homology with the SARS-CoV-1, [48, 53] and this homology has already been harnessed to predict epitopes based on experimental results from SARS-CoV-1. However, the different infection outcome of SARS-CoV-2 relative to SARS-CoV-1 originates in part from differences in the S protein. SARS-CoV-2 has better affinity to ACE 2 than SARS-CoV-1, yielding more efficient cellular entry [54, 55]. Furthermore, the presence of a furin cleavage site [56–58] in the S protein of SARS-CoV-2 (not present in SARS-CoV-1) coupled to an extended loop at the proteolytic site leads to higher cleavage efficacy thus facilitating its activation for membrane fusion [55, 59–61].

Analysis of 12 different plasma samples from SARS-CoV-2 infected patients and comparison to 6 samples from uninfected patients clearly highlighted a strong response to specific epitopes (Fig 2). The three linear epitopes most abundantly detected (SARS-CoV-2 S protein) were: 655–672, 787–822, and 1147–1158. None of these epitopes was singularly detected in all the positive samples tested, but each is detected in >40% of the positive patients. The 655–672 epitope is the most abundantly detected in positive samples and corresponds to a peptide that is not part of a secondary structures (Fig 3A and 3B). The corresponding epitope had been also detected in SARS-CoV-1 [8] (89% homology for the 18*mer* peptide, Fig 4A–4C) and predicted bioinformatically for SARS-CoV-2 [27, 31, 35, 36]; however, it had yet to be observed experimentally. Interestingly, this epitope is just next to the reported S1/S2 cleavage site (Fig 4A–4C, furin/TMPRSS2) [50, 57]. The proteolytic cleavage of the loop 681–685 has been demonstrated to be necessary for the viral entry into the host cell [50]. Moreover, the proteolytic cleavage of the S protein could be a determinant factor for the capacity of the virus to cross species. For example, the S protein of Uganda bats MERS-like CoV is capable of binding human cells, but this is insufficient for entry [62]. However, if a protease (trypsin) is added the protein is cleaved and viral entry occurs. Furthermore, the most closely related virus to SARS-CoV-2 is RaTG-13 from a bat found in Yunnan province in 2013 which does not contain the furin cleavage sequence [49]. Taken together, this evidence suggest that cleavage of the S protein is a barrier to zoonotic coronavirus transmission. Incorporation of the furin cleavage sites could have

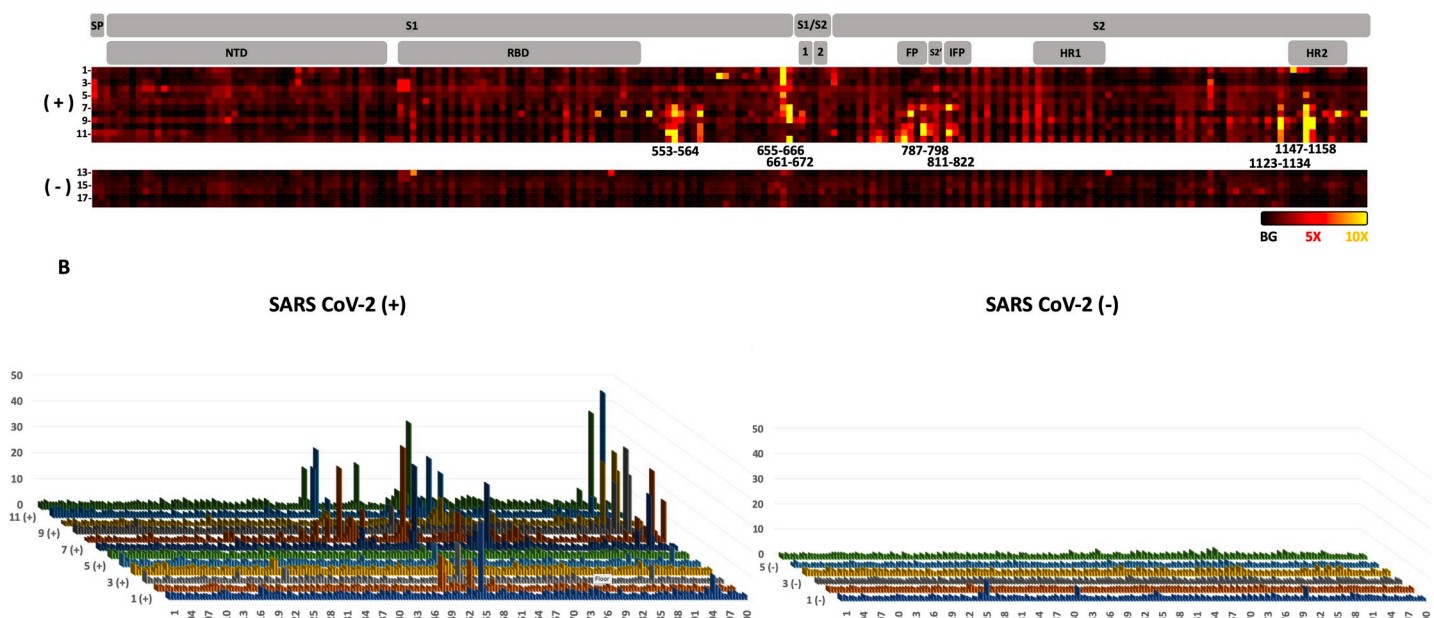

**Fig 2. Antibody response from 12 convalescent SARS-CoV-2 patients and 6 uninfected negative controls.** A: Domains of the spike protein (SP = Signal peptide, NTD = N-terminal domain, RBD = Receptor-binding domain, FP = Fusion Peptide, IFP = internal fusion protein, HR1 = Heptad repeat 1, HR2 = Heptad repeat 2) and heat map of antibody binding to the peptide fragments (black background intensity, red 5x background intensity and yellow 10x background intensity). Sample number are indicated on the left of the heat map. B: Fluorescence intensity of antibody binding from the 12 SARS-CoV-2 positive samples (left) and 6 SARS-CoV-2 negative samples (right). The fluorescence intensities are the median of 23 values followed by normalization to the background intensity. The immunodominant regions are highlighted with the corresponding residue numbers (the epitope numbers correspond to the column on top of the dash). See S1 Table in S1 File for quantification and summary of the data.

been acquired by recombination with another virus leading to human infection. In relation to the furin cleavage site, the pathogenic avian H5N1 contains such a furin cleavage site that leads to higher pathogenicity due to the distribution of furins in multiple tissues [63]. Most recently, high resolution structures analyzing the different conformation of the spike protein prior to and after furin-mediated proteolysis indicates that this proteolysis facilitates the conformational chage required for RBD exposure and binding to surface receptor [64]. We speculate that the binding of an antibody to the epitope 655–672 would sterically block the proteolysis of S1/S2 (*vide infra*) and should thus be broadly neutralizing, since this proteolysis is critical for infection.

Another epitope abundantly detected only in healing patients was the 787–822, a peptide segment extending at the periphery of the solvent exposed part of the protein (Fig 3A and 3B). It has also been experimentally observed in the SARS-CoV-1 [9, 13], SARS-CoV-2 [38, 39] and predicted bioinformatically [26, 27, 30, 31, 33, 36]. Interestingly, this epitope includes the S2' cleavage site of the spike protein (Fig 4D–4F), which has been reported to activate the protein for membrane fusion via extensive irreversible conformational changes [53, 65]. This epitope also includes the fusion peptide (816–833, Fig 4D–4F) [66] which is highly conserved among coronaviruses [67, 68], suggesting a potential pan-coronavirus epitope at this location. It should be noted that a peptide-based fusion inhibitor was shown to exhibit broad inhibitory activity across multiple human CoVs [69] and that antibodies against that region have shown neutralizing activity in SARS-CoV-1 [70]. Taken together, the data support the fact that antibodies inhibiting this proteolytic cleavage should be neutralizing [61, 66].

Finally, the epitope 1147–1158 is found at the C terminus of the spike protein. The structural data reported thus far did not suggest a defined structure for this portion of the S protein.

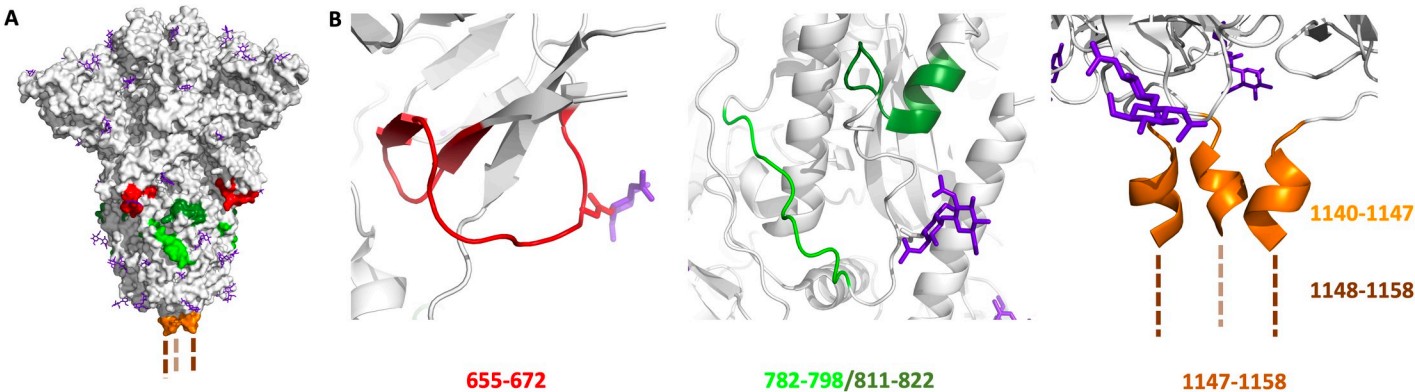

**Fig 3.** A) Localization of the three selected epitopes on the crystal structure of SARS-CoV-2 Spike protein (PDB ID: 6ZGE): red (epitope 655–672), green (epitope 782-798/811-822) and orange (epitope 1147–1158, the structure is undefined in the PDB). B) Expanded view of the 3 selected epitopes, *N*-linked glycan shown in purple.

This epitope extends from the helix bundle 1140–1147 (Fig 3A and 3B) and had also been experimentally observed in SARS-CoV-1 [9] and predicted bioinformatically for SARS-CoV-2 [27, 31, 35].

One limitation of epitope mapping with a peptide array is that it is restricted to linear epitopes. Antibodies binding to the RBD have been shown to participate in interactions spanning

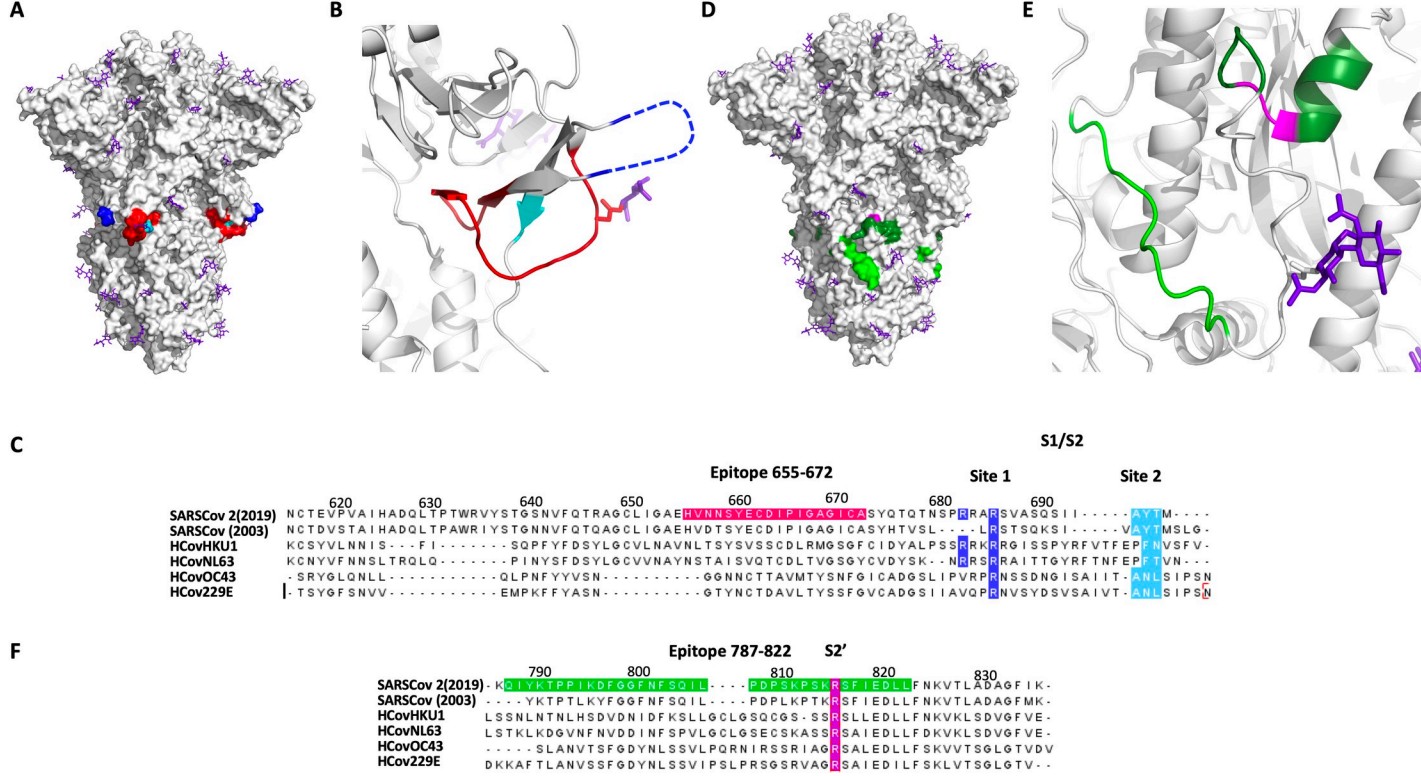

**Fig 4. Selected epitopes localization in relation to the protease cleavage site of the spike protein.** A-B) The 655–672 epitope (red) and the two reported protease cleavage sites S1/S2: site 1 (685–686: blue) and site 2 (695–696: cyano). C) Sequence alignment of the S1/S2 cleavage sites for five different coronaviruses SARSCov2 (2019), SARSCov (2003), HCovHKU1, HCovNL63, HCOVOC43 and HCov229E. D-E) The 787–822 epitope (green) and the S2' cleavage site (815–816: magenta). F) Sequence alignment of the S2' cleavage site. Figure generated from pdb ID: 6ZGE.

multiple peptide fragments. Indeed, we did not observe a strong response to linear peptides in the RBD. A control experiment with AI334/CR3022 antibody [25, 71] showed only weak binding to 367–378 peptide sequence of the RBD.

To validate the results observed on the microarray, a peptide (655–672) was synthesized as a biotin conjugate for pull-down and ELISA experiments. The sequence corresponding to 655–672-biotin and a scrambled version of the biotinylated peptide were individually immobilized on agarose streptavidin beads. Beads were exposed to serum from patients that were either positive or negative for that epitope based on the microarray data and subsequently treated with anti-Human-IgG-FITC. The fluorescence of the beads was quantified by confocal microscopy (Fig 5A). As can be seen in Fig 5B–5E, the beads with 655–672 peptide and positive serum sample showed higher fluorescence than the ones with either negative serum or using the scrambled peptide. To further probe the binding of 655–672 peptide to antibodies of SARS-CoV-2 positive patients, the same 655–672 biotinylated peptide was used in an ELISA assay (Fig 6A). Three SARS-CoV-2 positive samples showing strong 655–672 signal (Samples 7, 8 and 9) and three SARS-CoV-2 negative samples (Samples 14, 15 and 17) were analyzed showing clear binding to the 655–672 peptide and not to the scrambled version (Fig 6B).

Next, an alanine scan was performed to assess the contribution of individual amino acids to the interaction with the antibodies of two of the COVID positive patients containing antibodies for this epitope (Sample number 1 and 6) at two different dilutions (1 to 100 and 1 to 400). For this purpose, 17 different peptide-PNA conjugates were synthesized, replacing one amino acid at the time with Ala (Fig 7A) and measuring the intensity of the observed binding on the microarray. This analysis revealed the key role of 5 residues that, if converted to Ala, lead to dramatic loss of activity (amino acids in blue, Fig 7B). Thus, the key amino acids crucial for binding with the antibodies common for these two plasma samples are H655, Y660, C662,

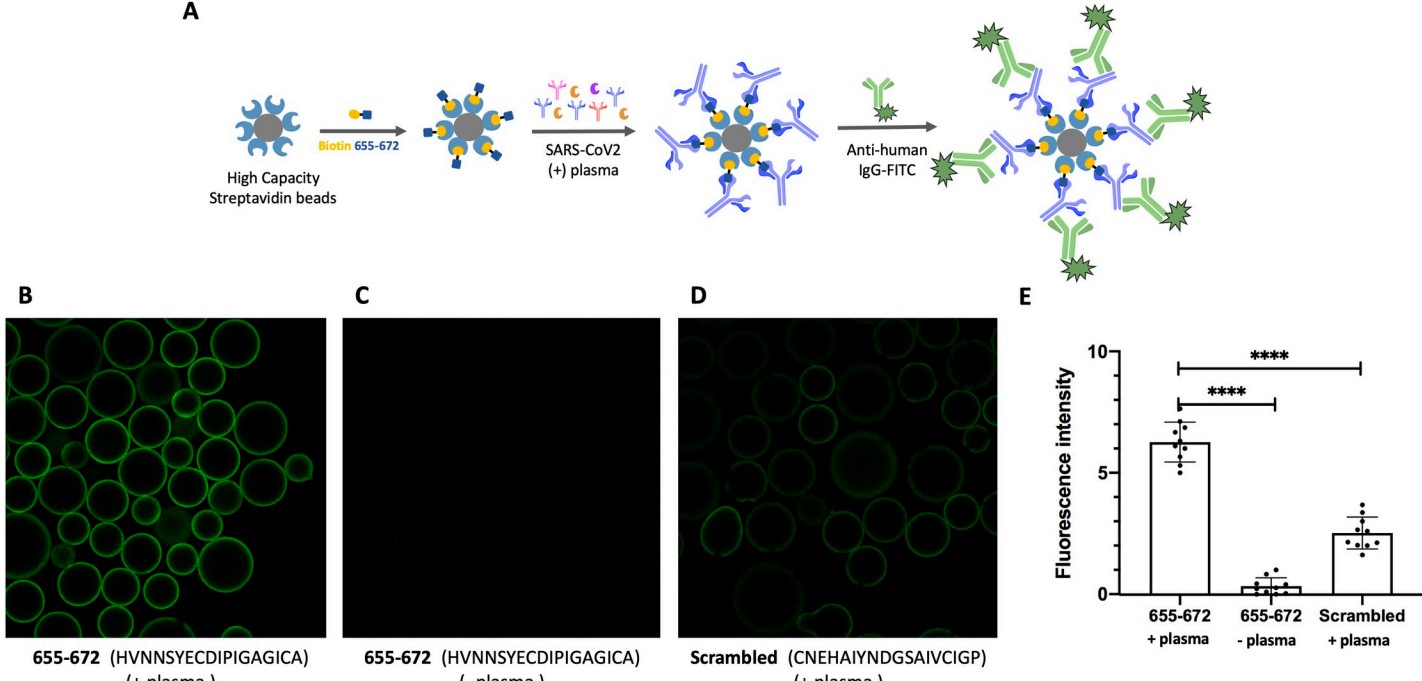

**Fig 5.** A) Schematic representation of epitope validation (anti-Spike-655-672 IgG in the SARS-CoV-2 positive patients' plasma). Microscope images of the beads with: B) Biotin 655–672 with Positive plasma; C) Biotin 655–672 with Negative plasma; D) Biotin-scrambled peptide with Positive plasma. E) FITC fluorescence quantification of B, C and D.

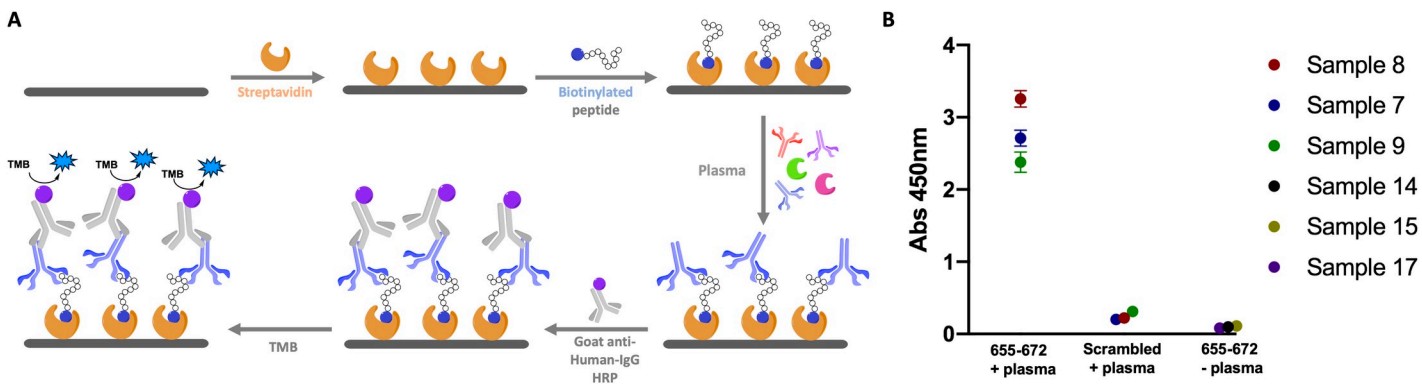

**Fig 6.** A) Schematic representation of the ELISA assay. B) ELISA assay with 3 different 655–672 positive samples with the 655–672 peptide and scrambled peptide and 3 negative samples. Error bars represent triplicate experiments.

G669 and C671. The binding of the antibody presents in sample 1 also seems to depend on the P665. The remarkable similarities between the two patients is notable considering a polyclonal response.

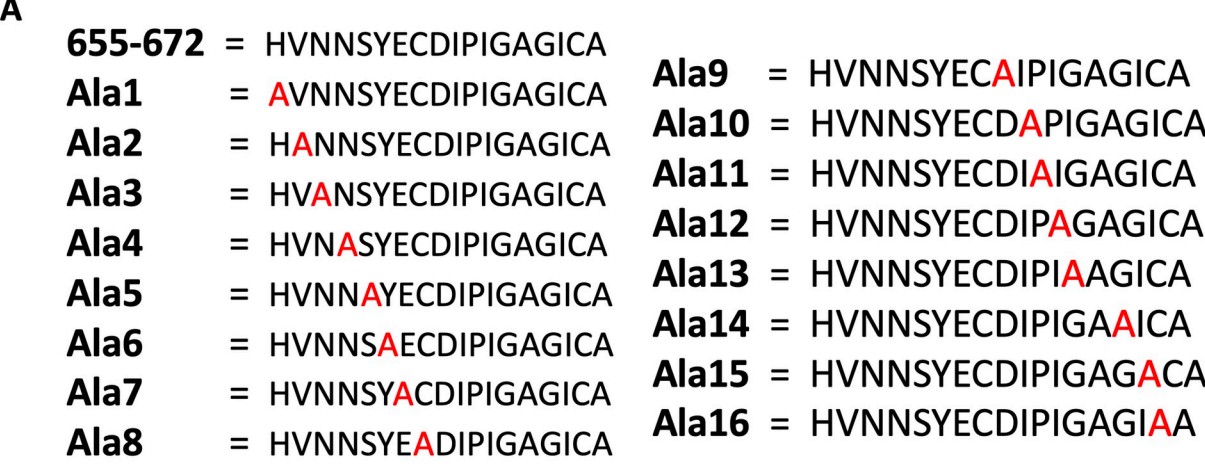

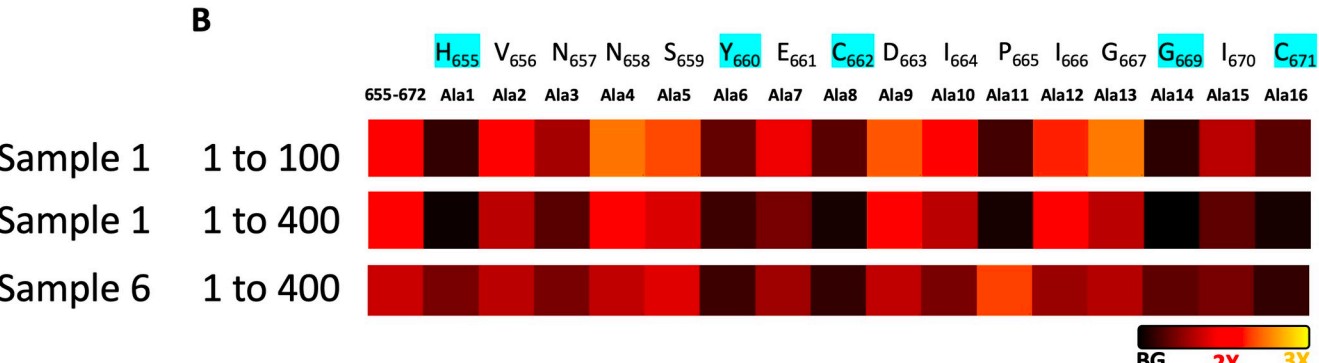

**Fig 7. Alanine scan by hybridization of PNA-peptide conjugates Ala-1 to Ala-17 to a DNA microarray.** A) Peptide sequences of the 17 PNA-peptide conjugates used for the alanine scan, where one amino acid at the time is modified by an alanine. B) Heat map of the interaction of 655–672 positive patient plasma with the different peptides-PNA conjugates hybridized in the DNA array at two different dilutions (1 to 100 and 1 to 400). Heat map represents the normalized fluoresce average of 23 different spots in the array.

It should be noted that the surface of SARS-CoV-2's spike is heavily *N*-glycosylated by host-derived glycans (22 *N*-glycosylation sites) with a potential role in camouflaging immunogenic protein epitopes [72]. Position N657, which is part of the identified epitope (655–672) adjacent to the furin cleavage site is glycosylated. The alanine scan indicated that this position does not contribute significantly to epitope-antibody interaction. A recently reported cryo EM structure with the 657-N-linked GlcNAc (6ZGE [64]) visible shows that the glycan point away from the epitope. Furthermore, a proteomic analysis showed that this position is unoccupied by a glycan in 16% of the monomers (i.e. nearly 50% of trimeric spike have an unoccupied N657) [72]. Glycosylation has also been reported at the edge of the other two prominent epitopes (N801, adjacent of epitope 787-798/811-822; N1158 at the edge of epitope 1147–1158). The fact that antibodies against these epitopes were observed in convalescing patients suggest that the glycans do not effectively shield access to these segments of spike. In the case of N801, the glycan projects away from the identified epitope [64]; in the case of N1158, there is no structural information available to date.

Based on the importance of the furin-mediated proteolysis, we next asked if plasma from patient positive for epitope 655–672 could protect the spike protein against proteolysis. To this end, the spike protein was labeled with Dylight 549 for visualization following SDS-PAGE. Treatment of labeled spike with furin for 20, 45 and 60 min afforded a progressive formation of two new lower molecular weight bands consistent with a single proteolytic event (one more intense band migrating at above the 70 KDa marker and a lower less intense band at *ca.* 70 KDa, S1 Fig in S1 File) and showed that 45 min was sufficient for nearly quantitative proteolysis under these conditions. Addition of plasma prior to the analysis does slightly alter the migration of the spike protein on the gel due to the increased protein loading however, the fluorescence scan still enabled a selective and unambiguous identification of the spike protein on the gel (S2 Fig in S1 File). Performing the proteolytic experiment with furin in the presence of plasma from a patient positive for epitope 655–672 (sample 12, Fig 2) showed a complete protection against proteolysis while plasma from a patient negative for this epitope (sample 10) did not (Fig 8). The comparison with a second sample from a patient positive for this epitope (sample 1) and the plasma from a healthy individual (sample 14) showed the same result (protection against proteolysis from sample 1 but not 14, S3 Fig in S1 File). These experiments support the fact that antibodies binding to epitope 655–672 are protective against furin-proteolysis of spike. This protection could be highly relevant in mitigating ADE by preventing viral entry irrespectively of antibody-mediated cellular interactions.

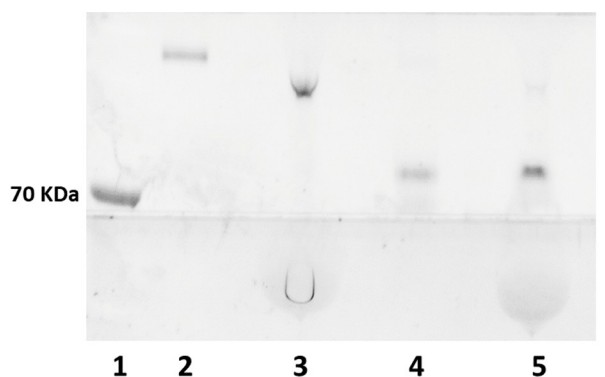

nCoV-19 spike-Dylight 549 $\xrightarrow{\text{furin}}$ S1 + S2

1: protein ladder
2: nCoV-19 spike-Dylight 549
3: nCoV-19 spike-Dylight 549 + plasma from sample 12 + furin
4: nCoV-19 spike-Dylight 549 + furin
5: nCoV-19 spike-Dylight 549 + plasma from sample 10 + furin

**Fig 8. Inhibition of furin-mediated proteolysis of spike.** Fluorescent scan of a SDS-PAGE with Dylight 549-labeled spike protein (lane 2) and treated with furin (lane 4). The same experiment was performed with the addition of plasma from patient 10 (negative for the epitope adjacent to the cleavage site, lane 5) and patient 12 (positive for the epitope adjacent to the cleavage site (lane 3). Lane 1 is a molecule weight marker.

## Conclusion

We have developed a peptide array for the epitope mapping of the spike protein of SARS-CoV-2. Using this array to profile healing plasma of twelve laboratory confirmed COVID-19 patients and six negative controls we have discovered three immunodominant linear regions, each present in >40% of COVID-19 patient (epitope 655–672 in 66%; epitope 782-798/811-822 in 40% and epitope 1147–1158 in 58%). Two of these epitopes correspond to key proteolytic sites on the spike protein (655–672: S1/S2 and 782-798/811-822: S2') which have been shown to be crucial for viral entry and play an important role in virus evolution and infection. We show biochemical evidence that serum positive for 655–672 epitope inhibits proteolysis of spike by furin. The fact that antibodies binding adjacent to the protease cleavage sites were identified from COVID-19 patients raises the possibility that other mechanism than blocking the RBD-ACE2 interaction could be harnessed for neutralization and might mitigate antibody-dependent enhancement of viral entry. Full characterization of these antibodies is necessary, and efforts on this direction are on their way.

## Supporting information

**S1 File.**
(PDF)

## Acknowledgments

The authors gratefully acknowledge Prof. Cosson from the Geneva Antibody Facility for generous gift of reagents, Isabelle Arm-Vernez from the virology laboratory of the laboratory medicine division for the assessment of routine SARS-CoV-2 IgG serology and Rémy Visentin from the Protein Facility at the Faculty of Medicine for assistance in recombinant spike protein production.

## Author Contributions

**Conceptualization:** Lluc Farrera-Soler, Sofia Barluenga, Nicolas Winssinger.

**Data curation:** Lluc Farrera-Soler, Jean-Pierre Daguer, Sofia Barluenga.

**Formal analysis:** Lluc Farrera-Soler, Jean-Pierre Daguer, Sofia Barluenga, Nicolas Winssinger.

**Funding acquisition:** Nicolas Winssinger.

**Investigation:** Lluc Farrera-Soler, Jean-Pierre Daguer, Sofia Barluenga, Nicolas Winssinger.

**Methodology:** Lluc Farrera-Soler, Jean-Pierre Daguer, Sofia Barluenga.

**Project administration:** Sofia Barluenga.

**Resources:** Oscar Vadas, Patrick Cohen, Sabrina Pagano, Sabine Yerly, Laurent Kaiser, Nicolas Vuilleumier.

**Supervision:** Sofia Barluenga, Nicolas Winssinger.

**Writing – original draft:** Lluc Farrera-Soler, Sofia Barluenga, Nicolas Winssinger.

**Writing – review & editing:** Lluc Farrera-Soler, Jean-Pierre Daguer, Sofia Barluenga, Oscar Vadas, Patrick Cohen, Sabrina Pagano, Sabine Yerly, Laurent Kaiser, Nicolas Vuilleumier, Nicolas Winssinger.

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
