## [Decision Letter · Decision Letter 0]

23 Jul 2020

PONE-D-20-18388

Identification of immunodominant linear epitopes from SARS-CoV-2 patient plasma

PLOS ONE

Dear Dr. Winssinger,

Thank you for submitting your manuscript to PLOS ONE. After careful consideration, we feel that it has merit but does not fully meet PLOS ONE’s publication criteria as it currently stands. Therefore, we invite you to submit a revised version of the manuscript that addresses the points raised during the review process.

The Reviews are somewhere between a major and minor revision, but readily addressable. The addition of neutralizing titers is important as is a summary table/figure as requested by Reviewer 2.

We look forward to receiving your revised manuscript.

Kind regards,

Nicholas J Mantis

Academic Editor

PLOS ONE

Journal Requirements:

Reviewers' comments:

Reviewer's Responses to Questions

**Comments to the Author**

1. Is the manuscript technically sound, and do the data support the conclusions?

Reviewer #1: Partly

Reviewer #2: Yes

2. Has the statistical analysis been performed appropriately and rigorously? 

Reviewer #1: Yes

Reviewer #2: No

3. Have the authors made all data underlying the findings in their manuscript fully available?

Reviewer #1: Yes

Reviewer #2: Yes

4. Is the manuscript presented in an intelligible fashion and written in standard English?

Reviewer #1: Yes

Reviewer #2: Yes

5. Review Comments to the Author

Reviewer #1: The manuscript entitled “Identification of immunodominant linear epitopes from SARS-CoV-2 patient plasma” describes a full scan of the linear epitopes of the spike of SARS-CoV- 2 with plasma from COVID19 patients collected 3 weeks after the PCR diagnosis. The scanning is done using 12-residue frames overlapping by 6 residues. The design of the experiments is standard and sufficiently reliable. The PNA attachment of the peptides to the microarray chip may produce neoepitopes sometimes but any microarray technique would create unnatural molecular context so this is an inevitable risk. The authors identify three epitopes, two of which overlap with cleavage sites crucial for the conformational change leading to fusion. The further conformation of the binding of patients’ plasma in different molecular context verifies that the actual interaction depends on the peptides themselves. Unfortunately, none of the validation tests addresses the actual binding to the native protein. Although, the hypothetical epitopes are selected on the basis of reactivity correlating with the disease, correlation is not causation. The necessity of native protein binding assay is dictated by the abundant glycosylation of the spike which occludes the greater part of the protein surface (according to the models reported by Hyeonuk Woo, et al. (2020)). The fact that p655-672 and p787-822 are adjacent to cleavage sites suggests that those parts of the polypeptide chain are sufficiently exposed. Still the conformational nature of the typical B cell epitopes requires that results of such linear epitope scanning be validated by a binding assay using the native protein. The logic that antibodies targeted away from the RBD and to the region of the cleavage site would prevent viral entry and, thus, ADE is contradicted in the case of the p655-672 by previous studies on SARS -CoV which showed that ADE is promoted by antibodies to the conserved epitope 611LYQDVNCT618 (the numbering is that of SARS-CoV-2) which is adjacent to the epitope p655-672. My humble opinion is that these contradictions need to be addressed in the final version of the paper.

Reviewer #2: This relatively straight forward report examines COVID-19 convalescent plasma for reactivity with peptide (12 mer) array spanning the major structural protein of SARS COV-2. The authors identify three linear epitopes reactive within a sample size of 12. The reactivities are validated by agarose bead, ELISA and alanine scanning. A major short coming of the study is that the convalescent plasma is not particularly well characterized for overall reactivity to SARS COV-2 or neutralizing potential. For that reason, the study is only of marginal significance.

Comments

1. The abstract is insufficiently detailed to be useful to the readers. The authors should refer to journal guidelines and expand on Methods, Results and Conclusions.

2. Figure 5E should ideally display individual data points within the bar graph

3. The manuscript is missing a single figure at the end that aggregate the results and provides a better sense of frequency of reactivity of the peptides. Statistical analysis should be included as well.

4. There is no mention of how glycosylation might affect linear epitope accessibility.

5. Finally, while the authors comment on viral neutralization, there the convalescent sera used in the study was not tittered for endpoint (EC50) or neutralizing activity (IC50). Both would have advanced the study considerably.

6. PLOS authors have the option to publish the peer review history of their article (what does this mean?). If published, this will include your full peer review and any attached files.

Reviewer #1: No

Reviewer #2: No

---

## [Author Response · Author response to Decision Letter 0]

30 Jul 2020

The response is provide in the cover letter. The text below is copied from the cover letter (without the formatting to distinguish reviewer comment from the response).

Dear Prof Mantis, dear editor,

Thank you for considering a revised version of the manuscript addressing the point raised by you and the reviewers. The most significant changes in the manuscript is the addition of biochemical evidence that plasma from patient positive for the epitope adjacent to the furin cleavage site does indeed inhibit the furin-mediated proteolysis of spike whereas plasma negative for this epitope does not. The second most important change is a discussion on the glycosylation sites; this discussion is aided by publications that appeared during the review process (new reference: 64 and 72). The frequency of epitope occurrence has been added to the conclusion and a table summarizing all data has been added (Table S1). Following your comment and the comment of reviewer 2, the plasma of two patient positive for epitope 655-672 were tested for neutralization and found to be neutralizing. However, we wish not to include this data because we cannot ascertain which antibody(ies) from the pool contribute(s) to the neutralization. As discussed in the manuscript, we acknowledge that the technique is limited to linear epitopes and might miss neutralizing antibodies directed at non-linear epitopes. One antibody targeting the receptor binding domain was reported to interact with multiple peptide across the RBD surface (Yan et al, Science 2020) and control experiments with this antibody only revealed weak binding, i.e. the presence of such antibody might be missed in the patient analysis. 

A point-by-point response is provided below. A marked version of the manuscript is submitted to facilitated the analysis of the changes. 

Review Comments to the Author

Reviewer #1: The manuscript entitled “Identification of immunodominant linear epitopes from SARS-CoV-2 patient plasma” describes a full scan of the linear epitopes of the spike of SARS-CoV- 2 with plasma from COVID19 patients collected 3 weeks after the PCR diagnosis. The scanning is done using 12-residue frames overlapping by 6 residues. The design of the experiments is standard and sufficiently reliable. The PNA attachment of the peptides to the microarray chip may produce neoepitopes sometimes but any microarray technique would create unnatural molecular context so this is an inevitable risk. The authors identify three epitopes, two of which overlap with cleavage sites crucial for the conformational change leading to fusion. The further conformation of the binding of patients’ plasma in different molecular context verifies that the actual interaction depends on the peptides themselves. Unfortunately, none of the validation tests addresses the actual binding to the native protein. Although, the hypothetical epitopes are selected on the basis of reactivity correlating with the disease, correlation is not causation. The necessity of native protein binding assay is dictated by the abundant glycosylation of the spike which occludes the greater part of the protein surface (according to the models reported by Hyeonuk Woo, et al. (2020)). The fact that p655-672 and p787-822 are adjacent to cleavage sites suggests that those parts of the polypeptide chain are sufficiently exposed. Still the conformational nature of the typical B cell epitopes requires that results of such linear epitope scanning be validated by a binding assay using the native protein. 

The point is well taken. Beyond the model proposed by Hyeonuk Woo et al 2020, experimental work characterizing the glycosylation pattern of spike has now been reported (Watanabe et al Science 2020; ref 71) as well as an EM structure showing the orientation of the glycan (Wrobel et al Nat. Struct. Mol. Biol. 2020; ref 64). The text has now been revised with a discussion including this new information. As detailed in the text, these glycosylation sites are not anticipated to preclude binding to the identified epitopes. Regarding the necessity of native protein binding assay, all tested samples tested positive in a ELISA assay against S1. To test the binding of the specific antibodies would require antibody purification which is not possible from the samples available. We are currently raising antibodies to these epitopes and the raised antibodies will clearly need to be validated for native protein binding, but this work extends beyond the scope of the present manuscript.

Text added to the manuscript: It should be noted that the surface of SARS-CoV-2’s spike is heavily N-glycosylated by host-derived glycans (22 N-glycosylation sites) with a potential role in camouflaging immunogenic protein epitopes.[72] Position N657, which is part of the identified epitope (655-672) adjacent to the furin cleavage site is glycosylated. The alanine scan indicated that this position does not contribute significantly to epitope-antibody interaction. A recently reported cryo EM structure with the 657-N-linked GlcNAc (6ZGE[64]) visible shows that the glycan point away from the epitope. Furthermore, a proteomic analysis showed that this position is unoccupied by a glycan in 16% of the monomers (i.e. nearly 50% of trimeric spike have an unoccupied N657) [72]. Glycosylation has also been reported at the edge of the other two prominent epitopes (N801, adjacent of epitope 787-798/811-822; N1158 at the edge of epitope 1147-1158). The fact that antibodies against these epitopes were observed in convalescing patients suggest that the glycans do not effectively shield access to these segments of spike. In the case of N801, the glycan projects away from the identified epitope [64]; in the case of N1158, there is no structural information available to date.

The logic that antibodies targeted away from the RBD and to the region of the cleavage site would prevent viral entry and, thus, ADE is contradicted in the case of the p655-672 by previous studies on SARS -CoV which showed that ADE is promoted by antibodies to the conserved epitope 611LYQDVNCT618 (the numbering is that of SARS-CoV-2) which is adjacent to the epitope p655-672. My humble opinion is that these contradictions need to be addressed in the final version of the paper.

We agree that the statement in the original version was confusing and the sentence has now been removed. The point that we wanted to make was that an antibody that would inhibit furin-mediated cleavage of spike might protect against ADE by preventing the required conformational changes to enable membrane fusion, irrespective of a binding-only antibody or RBD-targeted antibody redirecting the virus to FC-binding cells. This point is now strengthened by the recent publication (Wrobel et al Nat. Struct. Mol. Biol. 2020; ref 64) showing the importance of furin-mediated cleavage to achieve the conformation leading to high ACE2 affinity and membrane fusion. The point is further strengthened by the demonstration that plasma positive for epitope 655-672 inhibits cleavage whereas plasma negative for this epitope does not. Regarding the precedent for ADE mediated by 611LYQDVNCT618 in SARS-CoV-1, this epitope points away from the furin-cleavage site and would most likely result in binding-only antibodies. However, this discussion is immaterial since this epitope was not identified in the present study. 

Text added to the manuscript regarding the furin-protection assay: Based on the importance of the furin-mediated proteolysis, we next asked if plasma from patient positive for epitope 655-672 could protect the spike protein against proteolysis. To this end, the spike protein was labeled with Dylight 549 for visualization following SDS-PAGE. Treatment of labeled spike with furin for 20, 45 and 60 min afforded a progressive formation of two new lower molecular weight bands consistent with a single proteolytic event (one more intense band migrating at above the 70 KDa marker and a lower less intense band at ca. 70 KDa, Fig S1) and showed that 45 min was sufficient for nearly quantitative proteolysis under these conditions. Addition of plasma prior to the analysis does slightly alter the migration of the spike protein on the gel due to the increased protein loading however, the fluorescence scan still enabled a selective and unambiguous identification of the spike protein on the gel (Fig. S2). Performing the proteolytic experiment with furin in the presence of plasma from a patient positive for epitope 655-672 (sample 12, Figure 2) showed a complete protection against proteolysis while plasma from a patient negative for this epitope (sample 10) did not (Fig. 8). The comparison with a second sample from a patient positive for this epitope (sample 1) and the plasma from a healthy individual (sample 14) showed the same result (protection against proteolysis from sample 1 but not 14, Fig S3). These experiments support the fact that antibodies binding to epitope 655-672 are protective against furin-proteolysis of spike. This protection could be highly relevant in mitigating ADE by preventing viral entry irrespectively of antibody-mediated cellular interactions.

Reviewer #2: This relatively straight forward report examines COVID-19 convalescent plasma for reactivity with peptide (12 mer) array spanning the major structural protein of SARS COV-2. The authors identify three linear epitopes reactive within a sample size of 12. The reactivities are validated by agarose bead, ELISA and alanine scanning. A major short coming of the study is that the convalescent plasma is not particularly well characterized for overall reactivity to SARS COV-2 or neutralizing potential. For that reason, the study is only of marginal significance.

Comments

1. The abstract is insufficiently detailed to be useful to the readers. The authors should refer to journal guidelines and expand on Methods, Results and Conclusions.

We thank the reviewer for pointing out this shortcoming, the abstract has now been revised. 

2. Figure 5E should ideally display individual data points within the bar graph

Figure 5E has been revised to show individual data points.

3. The manuscript is missing a single figure at the end that aggregate the results and provides a better sense of frequency of reactivity of the peptides. Statistical analysis should be included as well.

A table summarizing all results as well as the quantification of microarray data has now been included (Table S1)

4. There is no mention of how glycosylation might affect linear epitope accessibility.

This is a good point that was also raised by reviewer 1 which was not discussed in the original submission for lack of robust experimental evidence regarding glycosylation pattern. Important new publications addressing this question have been published during the review process. This information is now part of the discussion (ref 64 for EM structure showing glycan and ref 72 for a proteomic analysis of the glycosylation pattern and frequency). 

5. Finally, while the authors comment on viral neutralization, there the convalescent sera used in the study was not tittered for endpoint (EC50) or neutralizing activity (IC50). Both would have advanced the study considerably.

The plasma from two patients positive for epitope 655-672 was tested for neutralization and found to be neutralizing (IC50) at 1:80 dilution. However, this information can be misleading and overinterpreted since we cannot be sure that the neutralization does not come from an antibody that is not detected using a linear epitope scan. As discussed in the manuscript, a control experiment with AI334/CR3022, an antibody that binds tightly to the RBD with interactions spanning multiple peptide fragments only gave weak binding using the linear peptide scan. In light of this information, any correlation between the antibody profile and neutralization activity may be misleading.

Thank you for your time and efforts during these challenging times, 

-Nicolas Winssinger

---

## [Decision Letter · Decision Letter 1]

11 Aug 2020

Identification of immunodominant linear epitopes from SARS-CoV-2 patient plasma

PONE-D-20-18388R1

Dear Dr. Winssinger,

We’re pleased to inform you that your manuscript has been judged scientifically suitable for publication and will be formally accepted for publication once it meets all outstanding technical requirements.

Kind regards,

Nicholas J Mantis

Academic Editor

PLOS ONE

Additional Editor Comments (optional):

Reviewers' comments:

Reviewer's Responses to Questions

**Comments to the Author**

1. If the authors have adequately addressed your comments raised in a previous round of review and you feel that this manuscript is now acceptable for publication, you may indicate that here to bypass the “Comments to the Author” section, enter your conflict of interest statement in the “Confidential to Editor” section, and submit your "Accept" recommendation.

Reviewer #1: All comments have been addressed

Reviewer #2: All comments have been addressed

2. Is the manuscript technically sound, and do the data support the conclusions?

Reviewer #1: Yes

Reviewer #2: Yes

3. Has the statistical analysis been performed appropriately and rigorously? 

Reviewer #1: Yes

Reviewer #2: Yes

4. Have the authors made all data underlying the findings in their manuscript fully available?

Reviewer #1: Yes

Reviewer #2: Yes

5. Is the manuscript presented in an intelligible fashion and written in standard English?

Reviewer #1: Yes

Reviewer #2: (No Response)

6. Review Comments to the Author

Reviewer #1: The additions are relevant and sufficient to address the previous concerns. I have no further comments.

Reviewer #2: My concerns have been addressed, although it would have been preferable if neutralizing activity could have been more central to the study.

7. PLOS authors have the option to publish the peer review history of their article (what does this mean?). If published, this will include your full peer review and any attached files.

Reviewer #1: **Yes: **Anastas Pashov

Reviewer #2: No

---

## [Editor Report · Acceptance letter]

18 Aug 2020

PONE-D-20-18388R1 

Identification of immunodominant linear epitopes from SARS-CoV-2 patient plasma 

Dear Dr. Winssinger:

I'm pleased to inform you that your manuscript has been deemed suitable for publication in PLOS ONE. Congratulations! Your manuscript is now with our production department. 

Kind regards, 

on behalf of

Dr. Nicholas J Mantis 

Academic Editor

PLOS ONE